# Public Perception on Healthcare Services: Evidence from Social Media Platforms in China

**DOI:** 10.3390/ijerph16071273

**Published:** 2019-04-10

**Authors:** Guangyu Hu, Xueyan Han, Huixuan Zhou, Yuanli Liu

**Affiliations:** School of Public Health, Chinese Academy of Medical Sciences and Peking Union Medical College, Beijing 100730, China; hugy@sph.pumc.edu.cn (G.H.); hanxueyan611@163.com (X.H.); chouhuixuan@live.cn (H.Z.)

**Keywords:** healthcare, social media, China, WeChat, Qzone, natural language processing

## Abstract

Social media has been used as data resource in a growing number of health-related research. The objectives of this study were to identify content volume and sentiment polarity of social media records relevant to healthcare services in China. A list of the key words of healthcare services were used to extract data from WeChat and Qzone, between June 2017 and September 2017. The data were put into a corpus, where content analyses were performed using Tencent natural language processing (NLP). The final corpus contained approximately 29 million records. Records on patient safety were the most frequently mentioned topic (approximately 8.73 million, 30.1% of the corpus), with the contents on humanistic care having received the least social media references (0.43 Million, 1.5%). Sentiment analyses showed 36.1%, 16.4%, and 47.4% of positive, neutral, and negative emotions, respectively. The doctor-patient relationship category had the highest proportion of negative contents (74.9%), followed by service efficiency (59.5%), and nursing service (53.0%). Neutral disposition was found to be the highest (30.4%) in the contents on appointment-booking services. This study added evidence to the magnitude and direction of public perceptions on healthcare services in China’s hospital and pointed to the possibility of monitoring healthcare service improvement, using readily available data in social media.

## 1. Introduction

Investigating public perception of healthcare services from different perspectives may generate inconsistent results. For example, patient-initiated violence against health workers [1,2,3,4,5,6], and the tension between doctors and patients for their dissatisfaction with the quality of healthcare [7,8], were wildly covered in the Chinese media. While patient experience surveys on the national level showed that patients were generally satisfied with both in-patient and out-patient services [9]. Such differences may result from biases rooted in the survey and media coverage; however, the inconsistency also pointed to the need for additional data sources to monitor public opinions on Chinese healthcare services.

It has been suggested that social media might be such a data source. Rozenblum et al. pointed out that when patient-centered healthcare, the internet, and social media were combined, the current relationship between healthcare providers and consumers might face major changes—thus creating a “perfect storm” [10]. Users’ posts on the social media platforms would generate a large volume of real-time data regarding public or private issues, among which healthcare related information scatters. Therefore, the utilization of social media data for healthcare research becomes a dramatically growing field and already covered various medical and healthcare research fields [11,12]. Sinnenberg and colleagues proposed four ways in which social media data were used in healthcare studies: (1) content analysis, (2) volume surveillance of contents on specific topics, (3) engagement of users with others, and (4) network analysis of users [12]. For the content analysis, most studies focused on measuring public discussion on specific diseases [13,14,15], sentiment analysis for medical interventions (e.g., cancer screening) [16,17], identifying safety concerns among health consumers [18], detecting adverse events of health products [19,20]. Several researchers studied patient experience, based on the comments posted by patients from online health communities in China [21,22], but few studies have been conducted to gather information on healthcare services related topics using social media data. Meanwhile, although sentiment analysis has been wildly applied to process user sentiments associated with health-related text [23], the lexical resource and tools designed for doing health-related sentiment analysis in Chinese language are few and far between.

Fast-advancing in technology and economy, social media users and their activities spiked in China, which made social media a promising source for healthcare service monitoring. In China, the internet penetration rate reached 55.8% at the end of 2017 [24], with local providers dominating the market, rather than Facebook and Twitter, which are not accessible in China. Chinese social media sites have a unique landscape, and it may not only be used as a communication software but also as an entry point for information. As subsidiaries of Tencent Holdings Limited, Shenzhen, China, WeChat and Qzone are two leading social media and networking services platforms. Each of them reached more than 938 million and 632 million monthly active user accounts in the first quarter of 2017 [25]. According to the 2016 WeChat Data Report, typical users of WeChat were born in the 80s or 90s [26], representing a wide breadth of demographic group in China. Besides providing multimedia communication and supporting social networking, WeChat also has “Official Accounts”, which serve as channels for publishing articles to the public. Any individual or organization can apply for having their own official account to broadcast their ideas and believes. As for Qzone, it is a platform bundled with QQ, a popular online messaging application in China. Qzone allows users to create their own personal page to write blogs and post updates. And users could be able to express their individual opinions and attitudes freely and instantly on the social media platforms. Subject to the platforms’ terms of service and privacy policy agreed upon by users [27,28], three kinds of information were collected, stored, and used by the platforms: (1) Personal information; (2) non-personal information; and (3) shared information. The shared information refers to information that is voluntarily shared on the platforms by users freely and instantly, thus providing a valuable perspective and opportunity to gather public opinions on healthcare services.

As such, we selected WeChat and Qzone as the social media platforms to conduct this exploratory study. The objectives of this study are to conduct volume and sentiment analyses base on the extracted social media contents on hospital healthcare services. The study could demonstrate the social media users’ perceptions of hospitals healthcare and may shed light on the further utilization of social media as a data source for healthcare research in China.

## 2. Materials and Methods

### 2.1. Study Design

This study consisted of three phases. Firstly, we utilized a predefined list of healthcare services categories to devise key words and search strategies accordingly. The data searching strategy would then be used to extract contents from a raw database, which contained publicized posts of WeChat and Qzone. The extracted materials were then put into a corpus. Secondly, we applied natural language processing (NLP) techniques from Tencent NLP platform to the corpus and calculated the volume of content concerning different healthcare services topics. Thirdly, we conducted sentiment analysis to explore the sentiment polarity of Chinese social media users on different healthcare service topics. The detailed process of data collection and analysis is presented in Figure 1. The study protocol was approved by the Ethics Committee of School of Public Health, Peking Union Medical College (71532014) and conducted under the academic collaborative project between Peking Union Medical College and Tencent.

### 2.2. Data Source

The raw databases used for this study come from WeChat and Qzone of the version only operated in mainland China. The user volumes and data inclusion criteria of the platforms were showed in Table 1. Publicly available posted information such as: posted blogs, reviews and articles that are voluntarily shared by individual users from June 2017 to September 2017 were collected from the two platforms. The data collection followed the privacy policy for users of Tencent and was subject to the confidentiality and security measures that implemented by the platforms. And the data analyses were supported by technicians in the Tencent.

### 2.3. Healthcare Services Categories

The nine healthcare service categories, used in this study, were derived from the objectives of the National Healthcare Service Improvement Initiative (2015–2017), which was dedicated to improving patient-centered healthcare and patient experience nationwide by the former National Health and Family Planning Commission of P.R. China (NHFPC) [29]. The initiative operated under the leadership of the Bureau of Medical Administration of NHFPC [30], which suggested that we used nine predefined categories to reflect the healthcare services in hospital (see Table 2). 

### 2.4. Searching Strategy and Corpus Construction

In this study, we constructed a healthcare services corpus, in the Chinese language from the social media data source, to enable further analyses. First, we constructed lexica of keywords and terms in accordance with the predefined service topics. For example, the lexicon for “Information technology”, used in this study indicate new information dissemination channels, based on information technology provided by hospital to improve patient experience of service information acquisition. And this lexicon contains six information technology service-related terms, namely, “Weibo”, “WeChat”, “website”, as well as “Self-service machine”. Second, we developed a set of searching strategy to extract the relevant data from the two sources based on the corresponding lexicon of topics. The entire list of search terms for each category and its corresponding searching strategy were provided in Appendix A. Finally, we applied the search strategies to the database of publicly posted materials to screen for posts related to the healthcare service categories to construct the corpus. The search and screening process were performed by Qcloud.

### 2.5. Analyze the Social Media Content of Healthcare Services

Based on the healthcare services corpus, we classified the content to different healthcare services topics that predefined and measured the content volume of the topic. Specifically, we used the open application programming interface (OpenAPI) services provided by Tencent NLP to analyze the retrieved contents. It is an open platform for Chinese natural language processing (based on parallel computing and distributed crawling system) [31]. Such services enable us to split reviews and blogs into sentences, and each sentence was filtered to classify whether it contained target service topic keywords and terms. If the sentences, containing certain keywords and terms, belonged to the corresponding topic of healthcare services categories as listed in Appendix A, then they would be divided into a certain category. By counting the appearances of each service topic keywords in terms of the number of sentences in the corpus, we can aggregate the counts at the topic level and calculate the proportion of different topics from the social media corpus. For the sentiment analysis tool in Chinese, we also select Tencent NLP, as its algorithm was trained by hundreds of billions of entries of internet corpus data in Chinese and with successful application in other Tencent products (https://nlp.qq.com). OpenAPI with function of Chinese batch texts automatic summarization and sentiment analysis of Tencent NLP enable us to categorize the sentences on certain topic in the social media corpus into a sentiment polarity classification (i.e., neutral, positive, and negative). Finally, each sentence was tagged and classified into different sentiment polarity.

## 3. Results

### 3.1. Content Volume

The social media corpus contained approximately 29 million records from WeChat and Qzone, spanning the 9 pre-defined categories, related to hospital healthcare services.

Table 3 presents the content volume of each healthcare services topic by social media channel. Among the social media content on healthcare services topics, patient safety was the most commonly encountered topic, both in WeChat and Qzone. The majority of the content related to patient safety issue, its approximately 8.73 million records and covered 30.1% of the entire corpus. The proportion of contents related to other topics varied in the corpus: Information technology (22.2%), service efficiency (17.9%), service environment (10.3%), inpatient service (9.6%), appointment-booking service (3.4%), nursing service (2.5%), doctor-patient relationship (2.5%), and humanistic care (1.5%).

### 3.2. Sentiment Analysis

The results of the sentiment analysis of contents from the corpus found that, in all nine healthcare services topics, 36.1% of the contents in the corpus have been recognized to reveal a positive disposition, 16.4% neutral and 47.4% negative. We found that topic comprising most positive contents was service environment (59.6%), followed by patient safety (53.2%). With regard to the topics that contained more negative contents than positive, the most one was doctor-patient relationship (74.9%), followed by service efficiency (59.5%), and nursing service (53.0%). Notably, over one third of contents in the appointment-booking service (30.4%) revealed a neutral disposition. 

Additionally, in contrast to the content volume distribution for the nine topics, the sentiment disposition of contents in corresponding healthcare services topics shows differences. For instance, Table 3 shows that the nursing service and doctor-patient relationship share an equal proportion (2.5%) of contents in the corpus, however, we observed the disposition of contents from social media users to the two topics varied in Figure 2.

## 4. Discussion

To our knowledge, this is the first study that has attempted to explore the public perceptions of healthcare services, using publicly posted materials, of two Chinese social media platforms. Our results showed that patient safety was the most significant topic for users of Chinese social media platforms, followed by information technology and service efficiency. Service environment was found to have the highest proportion of positive comments.

The research assessed the application of content volume calculation and sentiment analyses on Chinese social media data. The study is a crucial step to discovering the methodology on harnessing the social media data in China and an early attempt to track the perceptions of healthcare services in the public by analyzing a unique data source.

This study found a large number of information technology and service efficiency, which might reflect the series of efforts made by both the government and the hospital in integrating information technology in healthcare services in China. Several researchers have identified that health information technology services were used to enhance patient experience [32,33,34], and as a potential solution to shorten the lengthy waiting time in China’s public hospital [22,35,36,37].

Humanistic care was the least mentioned topic in the corpus complied by this study. It may suggest that Chinese social media users are not very familiar with the idea of humanistic care. Those who posted about it basically expressed a positive attitude. An alternative explanation might be this type of care has yet to reach the public only experienced by a few people. Further empirical studies or controlled studies may be conducted to provide further insights.

Our research also explored the sentiment disposition of social media content on healthcare services: 47.4% provided negative feedback. Although this was only the initial results, it could be quite alarming to healthcare administrations and policymakers. Despite the fact that patient surveys generally had favorable results in China [9], there was still a significant amount of negative comments on the social media platforms. Further and more detailed methodology is necessary to further understand the negative comments.

In the 9 topics investigated in this study, we found huge variations in the negative feedback as well as content volumes across topics. For instance, the contents related to doctor-patient relationship only take percentage of 2.5% in the corpus, however 74.9% of the content revealed negative feedback. The varied sentiment polarity distribution of the topics may have important policy implications for healthcare reform in China. For example, 30.4% of the social media references to appointment-booking service reflected neutral feedback, which may suggest that the unsureness of the public on this novel service. Patients have yet to be familiar with the services—even though it certainly aims to improve the convenience for patients as well as hospital efficiency. Such feedback could be essential for hospitals to improve their service quality by enhancing patient education. Further research might focus on what exactly were discussed in those negative posts so that targeted measures can be employed by the hospitals and responsible administrators to improve the services.

In line with previous evidence [11,12], our results show that social media could be a useful tool for health research in China, as well as English, and could be used to capture the public’s perspective of healthcare [23,38]. However, it appeared that the most concerned issue of healthcare in social media is different from what has been found in patient surveys. Findings from a recent qualitative study found that patients cared about the environment and facilities in hospital the most [39], whereas in our study patient safety issues had the greatest volume. Another research examined the online doctor reviews in China revealed that most posts expressed positive attitudes towards the physicians [21]. Although the evidence on these issues are still not conclusive, it might suggest the perception difference between general public and patients.

## 5. Strengths and Limitations

Our research extends application of the natural language processing techniques to analysis of healthcare services related contents in China’s social medial platforms and offers a new perspective of healthcare services in China’s hospital. The results would be of benefit to healthcare providers and regulators benchmarking their performance on patient-centered healthcare delivery. This is important because the social media has been considered as a portal of health information acquisition for Chinese netizens [40], the perspective of social media would be supplementary in understanding how consumer views the healthcare services in hospital besides the results from traditional paper-based surveys.

This study has the following limitations. First, because the raw material was user-generated data, selection bias may have affected the data. For example, it was observed that most social media users were born in the 80 s or 90 s [26], however we were unable to characterize the users social-demographic information in detail, since the user privacy policy of Tencent currently prohibit such practice. Moreover, considering the exploratory nature of the study, our study focused only on WeChat and Qzone as data sources, whereas other social media platforms in China may have the potential to conduct such analyses as well.

Second, since we derived the healthcare services categories and lexica based on the government document on NHSII and expert consultation, thus the corpus in this study may have failed to include certain amount of healthcare services related data. As a result, we may have underestimated the content volume of healthcare services from the two social media platforms. Furthermore, although all the material in the databases are in Chinese, and therefore most likely be generated by users from China, we are currently not able to determine whether the posts, containing the key terms on healthcare, were describing the Chinese healthcare system or discussing foreign healthcare systems in Chinese language. Further research may strive to develop searching strategies that enable such distinction and increase the specificity of the results.

Third, although the consumer health vocabulary (CHV) is the gold standard reference for retrieving the target data, it has been used in previous researches [13,38], such open source of vocabulary list and its corresponding lexica are not available in Chinese language. The accuracy and credibility of the sentiment analysis of this study also await further validation; however, it would require an alternative method to conduct sentiment analyses for Chinese language and the possibility to apply such methods on the Tencent data, which were publicly posted material but still under strict terms of utilization. Another limitation concerned that we have no ability to confirm that the data supplied by Tencent completely represent all users’ data as there could be undocumented keyword filter on the platforms. These would inflict potential bias and limit the generalizability of our findings.

## 6. Further Research

Weibo is another popular Chinese social media platform considered to be the counterpart of Twitter in China. Future research could consider extend the analysis process to contents from Weibo, to further explore users, and their views, that have not been covered in this study.

Both the quantitative approach, as shown in our research, and the qualitative approach, such as the face-to-face individual interview method, would be useful to better understand consumer care in healthcare services. There is a scarcity of empirical research exploring the latter issue at present. It has been proposed to complement public perspectives on healthcare services [39].

Furthermore, the popularity of consumers’ unsolicited comments on healthcare providers in social media, prompts an important avenue for understanding patient experience, and has been demonstrated by previous researches [38,41,42,43,44]. Future research for measuring patient experience based on social media data at hospital level would be help to better understand the landscape of healthcare quality in China.

## 7. Conclusions

By analyzing shared information from WeChat and Qzone, this study showed that patient safety was the most concerned topic for users of Chinese social media platform, followed by information technology and service efficiency, while the doctor-patient relationship was found to have the highest proportion of negative comments. This study explored the possibility of utilizing social media to monitor public perceptions on healthcare services. The findings provide an overview of public opinion on healthcare services, which could help regulators to set up the benchmark, on a national or regional level, to monitor the progress of healthcare improvements between comparator districts and services domains. It is also a necessary complement to the traditional paper-based consumer survey. The potential differences between social media perception and traditional consumer survey results would help regulators better understand the gap in quality of care services from various perspectives. Further studies could also focus on extending the NLP method to a more content-based resource and to expand our understanding of mass opinion on healthcare services.

## Figures and Tables

**Figure 1 ijerph-16-01273-f001:**
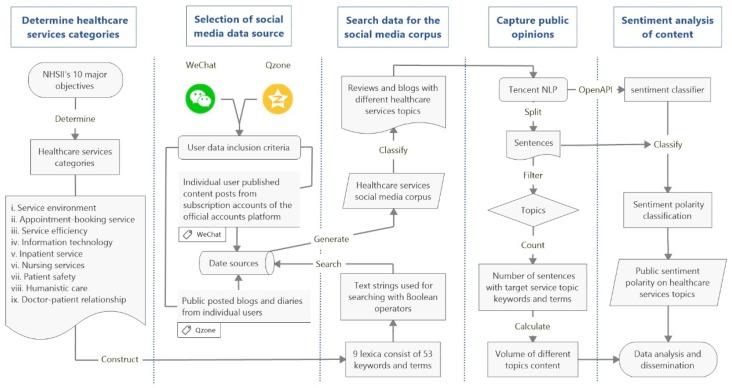
Flowchart of data collection and analysis. Note: NLP, natural language processing; API, application programming interface.

**Figure 2 ijerph-16-01273-f002:**
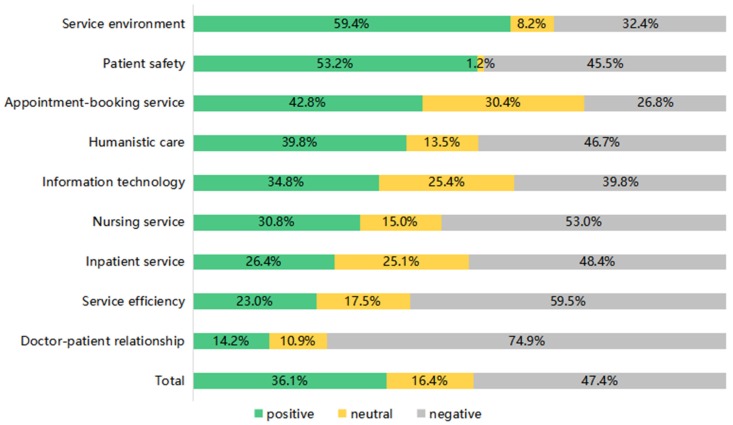
Comparison of sentiment expressed towards the healthcare services topics at overall level.

**Table 1 ijerph-16-01273-t001:** Data sources.

Platform	Launch Time	Monthly Active User Accounts (2017Q1)	User Data Inclusion Criteria
WeChat	2011	938 million	Posts from subscription accounts of the official accounts published by users
Qzone	2005	632 million	Public posted blogs and diaries of individual users

**Table 2 ijerph-16-01273-t002:** Healthcare services categories and corresponding descriptions in the National Healthcare Service Improvement Initiative 2015–2017 (NHSII).

Healthcare Services Categories	Objectives of NHSII
Service environment	Optimize the layout of the facility and build a friendly service environment
Appointment-booking service	Promote utilization of clinical appointment services and guide patient flow
Service efficiency	Improve service efficiency and effectiveness by rational allocation of resources
Information technology	Take advantage of information technology to improve patient experience
Inpatient service	Promote inpatient service process reengineering and provide integrated healthcare service
Nursing service	Continuously improve quality of nursing care and enhance nursing workforce
Patient safety	Ensure patient safety by promoting adoption of standard operating procedures
Humanistic care	Strengthen humanistic care and provide medical social worker service
Doctor-patient relationship	Harmonize the doctor-patient relationship and reduce medical disputes

**Table 3 ijerph-16-01273-t003:** Distribution of each topic content volume (count and proportion), 15 June 2017 to 15 September 2017.

Healthcare Services Topic	WeChat (*N* = 15,172,421)		Qzone (*N* = 13,844,634)		Total (*N* = 29,017,055)	
	Count	%	Count	%	Count	%
Patient safety	4,020,928	26.5%	4,704,284	34.0%	8,725,212	30.1%
Information technology	3,598,566	23.7%	2,857,266	20.6%	6,455,832	22.2%
Service efficiency	2,491,950	16.4%	2,703,352	19.5%	5,195,302	17.9%
Service environment	1,902,727	12.5%	1,075,697	7.8%	2,978,424	10.3%
Inpatient service	1,392,611	9.2%	1,396,209	10.1%	2,788,820	9.6%
Appointment-booking service	641,865	4.2%	353,856	2.6%	995,721	3.4%
Nursing service	424,229	2.8%	303,034	2.2%	727,263	2.5%
Doctor-patient relationship	438,823	2.9%	276,865	2.0%	715,688	2.5%
Humanistic care	260,722	1.7%	174,071	1.3%	434,793	1.5%

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
