# Peer review of "Public Perception on Healthcare Services: Evidence from Social Media Platforms in China"

_ijerph, 2019, doi:10.3390/ijerph16071273_

Round 1
Reviewer 1 Report
This is an interesting article demonstrating the potential that social media has to help the improvement in our understanding of population's and their disposition to healthcare. The design and methodology used appear appropriate but I do have some concern upon the reliance in using the social media company to provide the data, which I imagine was de-identified other than age range which I presume is why you are able to state that the responders were born in the 1980s and 1990s? It would have been helpful, if you had access to any demographic data, to give the age ranges, gender and area of residence (urban/rural) to allow for further depth of understanding. For clarity it might have been helpful to have the supplementary information in English as well as Chinese. The findings are presented clearly but seem to stop without suggesting potential healthcare impact of the 9 healthcare service categories. As suggested further analysis of the data set is needed - maybe another article? - and further discussion around the impact to healthcare service provision.
Author Response
Response to Reviewer 1 Comments
Point 1: The design and methodology used appear appropriate but I do have some concern upon the reliance in using the social media company to provide the data, which I imagine was de-identified other than age range which I presume is why you are able to state that the responders were born in the 1980s and 1990s?
Response 1: Thanks for this important feedback. The data was indeed de-identified and we reported the typical users of WeChat were born in the 80s or 90s (line 63) is quoted from the 2016 WeChat Data Report as a published reference (reference 26) rather than from our research. Nevertheless, we modified the sentence as follows to address any unclarities:
“According to the 2016 WeChat Data Report, typical users of WeChat were born in the 80s or 90s [26], representing a wide breadth of demographic group in China.” (line 62-63)
Point 2: It would have been helpful, if you had access to any demographic data, to give the age ranges, gender and area of residence (urban/rural) to allow for further depth of understanding.
Response 2: Thank for this suggestion, but we are unable to access any demographic data since the internal policy of Tencent currently prohibit such user profiling (line 93-94). As mentioned in the manuscript, three kinds of information were collected, stored and used by the platforms: 1) personal information, 2) non-personal Information and 3) shared information, among them only shared information could be accessed. In accordance with the reviewer concerns, we revised the related content in the part of limitation as follows:
“First, because the raw material of the study was user-generated data, there might be selection bias in the data we analyzed. For example, it was observed that the majority of social media users born in the 80s or 90s [26], however we were unable to characterize the users social-demographic information in detail since the user privacy policy of Tencent current prohibit such practice.” (line 226-230)
Point 3: For clarity it might have been helpful to have the supplementary information in English as well as Chinese.
Response 3: We are grateful for the suggestion and have added the English version of the supplementary information in the appendix.
Point 4: The findings are presented clearly but seem to stop without suggesting potential healthcare impact of the 9 healthcare service categories. As suggested further analysis of the data set is needed - maybe another article? - and further discussion around the impact to healthcare service provision.
Response 4: Regarding the suggestion on further analysis, we further revised the discussion session as follows:
“Humanistic care was the least mentioned topic in the corpus complied by this study. It may suggest that Chinese social media users are not very familiar with the idea of humanistic care. The ones who actually posted about it basically expressed a positive attitude. An alternative explanation might be this type of care has yet to reach the public only experienced by a few people. Further empirical studies or controlled studies may be conducted to provide further insights.” (line 185-189)
“The varied sentiment polarity distribution of the topics may have important policy implications for healthcare reform in China. For example, 30.4% of the social media references to appointment-booking service reflected neutral emotion, which may suggest that the unsureness of the public on this novel service. Maybe the patients have yet to be familiar with the service—despite the fact that it certainly aims at improving convenience for patients and hospital efficiency—and such feedback could be essential for hospitals to improve their service quality by enhancing patient education. Further research might focus on what exactly were discussed in those negative posts so that targeted measures can be employed by the hospitals and responsible administrators to improve the services.” (line 198-206)
Reviewer 2 Report
Dear Colleagues, Many thanks for the opportunity to review this interesting manuscript addressing the expressed attitude of service users, with regard to mental health provision, through sentiment analysis of a corpus assessembled from online social media platforms. I found myself agreeing with the authors as to the significance of such media as a means of communication around concepts of mental health and service provision. However, I also had a number of concerns regarding the proposed methodology and the depth of the discussion in relation to the emergent findings. I consider these points, and others below. In the introduction to the paper I would have welcomed a more detailed general introduction to the field of online social media, as accessed within China. A more detailed introduction to the two media platforms analysed would have been welcomed. I also wondered about the international reach of these platforms : Can we be certain, outside of using geolocation data, as to which particular mental health services are being referred to i.e. within or without China? In the methodology section - I would have welcomed a more detailed explanation as to exactly how the corpus for analysis was assembled? I found it difficult to determine precisely how it was recognised the posts, on the social media platforms, related to issues of mental health? In dividing the corpus into the categorical headings (humanistic care etc) again I was unclear as to precisely how this was completed? For example, were these exact headings used in an analysis of the substrate text, or were synonyms also employed? The description of the sentiment analysis stage was also very limited - please could more details be provided? In addition, were any verification stages employed to assess the accuracy of the sentiment analysis? Research addressing machine led sentiment analysis for an English language corpus often find errors, or misunderstanding, of sentiment - particularly when humour (sarcasm) is used by the author. In the discussion section, I would have welcomed a wider consideration of the presented material in relation to recognised issues for mental healthcare in China and around the World? For example, how do the authors see this work as relating to similar work conducted in other countries or regions of China? Overall, I find myself agreeing with the authors as to the significance of such research. However, given the intended International audience of the journal I wondered whether a greater global context could be applied to the findings? Finally, I felt that the manuscript would have benefitted from a close reading in terms of language accuracy as at times the meaning of utterances and claims risked being lost. Kind regards
Author Response
Point 1: In the introduction to the paper I would have welcomed a more detailed general introduction to the field of online social media, as accessed within China. A more detailed introduction to the two media platforms analysed would have been welcomed. I also wondered about the international reach of these platforms: Can we be certain, outside of using geolocation data, as to which particular mental health services are being referred to i.e. within or without China?
Response 1: Thanks for this important feedback. To be clearer to the reviewer concerns, we added general introduction information as follows:
“With fast advancing in technology and economy, the social media users and their activities spiked in China which made social media a promising source for healthcare service monitoring. In China, the internet penetration rate has reached 55.8% by the end of 2017 [24], with local providers dominate the market rather than Facebook and Twitter, which are not accessible in China. Chinese social media sites have a unique landscape, and it may not only be used as a communication software but also as an entry point for information.” (line 54-59)
“Besides providing multimedia communication and supporting social networking, WeChat also has “Official Accounts”, which serve as channels for publishing articles to the public. Any individual or organization can apply for having their own official account to broadcast their ideas and believes. As for Qzone, it is a platform bundled with QQ, a popular online messaging application in China. Qzone allows users to create their own personal page to write blogs and post updates. And users could be able to express their individual opinions and attitudes freely and instantly on the social media platforms.” (line 63-69)
We also add the limitation section as follows:
“Furthermore, although all the material in the databases are in Chinese and therefore most likely be generated by users from China, we are currently not able to determine whether the posts contained the key terms on healthcare were describing the Chinese healthcare system instead of describing or discussing foreign healthcare systems in Chinese language. Further research may strive to develop searching strategies that enable such distinction and increase the specificity of the results.” (line 236-241)
Point 2: In the methodology section - I would have welcomed a more detailed explanation as to exactly how the corpus for analysis was assembled?
Response 2: Thanks for this suggestion. The corpus was assembled according the flowchart we have displayed in Figure 1 and detailed process were reported in section 2.4 with revision as follows:
“First, we constructed lexica of keywords and terms in accordance with the predefined service topics. For example, the lexicon for “Information technology” used in this study indicate new information dissemination channels based on information technology provided by hospital to improve patient experience of service information acquisition. And this lexicon contains six information technology service-related terms, namely, “Weibo”, “WeChat”, “website”, “Self-service machine”, etc. Second, we developed a set of searching strategy to extract the relevant data from the two sources based on the corresponding lexicon of topics. The entire list of search terms for each category and its corresponding searching strategy were provided in Supplementary Table S1. Finally, we applied the search strategies to the database of publicly posted materials to screen for posts related to the healthcare service categories to construct the corpus. The search and screening process were performed by Qcloud.” (line 113-123)
Point 3:I found it difficult to determine precisely how it was recognised the posts, on the social media platforms, related to issues of mental health?
Response 3: The data for analysis come from all the publicly posted information such as: posted blogs, reviews and articles that are voluntarily shared by individual from the social media platforms in a database, and then we applied the predefined healthcare service searching strategies to the materials in the database as described in section 2.4, thus the posts concerned the issues of interest were recognized.
Point 4:In dividing the corpus into the categorical headings (humanistic care etc) again I was unclear as to precisely how this was completed? For example, were these exact headings used in an analysis of the substrate text, or were synonyms also employed?
Response 4: Thanks for this important feedback. To be clearer to the reviewer concerns, we revised the relevant content in section 2.5 as follows:
“Such services enable us to split reviews and blogs into sentences, and each sentence was filtered to classify whether or not it contained target service topic keywords and terms. If the sentences contained certain keywords and terms belonged to the corresponding topic of healthcare services categories as listed in Table S1, then they would be divided into a certain category.” (line 59-61)
Point 5:The description of the sentiment analysis stage was also very limited - please could more details be provided? In addition, were any verification stages employed to assess the accuracy of the sentiment analysis? Research addressing machine led sentiment analysis for an English language corpus often find errors, or misunderstanding, of sentiment - particularly when humour (sarcasm) is used by the author.
Response 5: Thank you for underlining this deficiency. The detailed information about sentiment analysis added as follows:
“For the sentiment analysis tool in Chinese, we also select Tencent NLP, due to its algorithm was trained by hundreds of billions entries of internet corpus data in Chinese and with successful application in other Tencent products (https://nlp.qq.com). OpenAPI with function of Chinese batch texts automatic summarization and sentiment analysis of Tencent NLP enable us to categorize the sentences on certain topic in the social media corpus into a sentiment polarity classification (i.e., neutral, positive, and negative). Finally, each sentence was tagged and classified into different sentiment polarity.” (line 135-141)
For the verification issue, we revised the limitation section according to the comment as follows:
“The accuracy and credibility of the sentiment analysis of this study also await further validation; however it would require an alternative method to conduct sentiment analyses for Chinese language and the possibility to apply such methods on the Tencent data, which were publicly posted material but still under strict terms of utilization.” (line 244-248)
Point 6:In the discussion section, I would have welcomed a wider consideration of the presented material in relation to recognised issues for mental healthcare in China and around the World? For example, how do the authors see this work as relating to similar work conducted in other countries or regions of China?
Response 6: Thanks for this suggestion. We complemented the discussion section as follows:
“In line with previous evidence [11, 12], our results show that social media could be a useful tool for health research in Chinese as well as English, and could be used to capture the public’s perspective of healthcare [23, 39]. However, it appeared that the most concerned issue of healthcare in social media is different from what has been found in patient surveys. Findings from a recent qualitative study found that patients cared about the environment and facilities in hospital the most [40], whereas in our study patient safety issues had the greatest volume. Another research examined the online doctor reviews in China revealed that most posts expressed positive attitudes towards the physicians [21]. Although the evidence on these issues are still not conclusive, it might suggest the perception difference between general public and patients.” (line 207-216)
Point 7:Overall, I find myself agreeing with the authors as to the significance of such research. However, given the intended International audience of the journal I wondered whether a greater global context could be applied to the findings?
Response 7: Thanks for this important feedback. We have added the conclusion section as follows:
“The finding provides an overview of public opinion on healthcare service, this could help the regulators of authority set up the benchmark on national or regional level to monitor the progress of healthcare improvement between comparator districts and services domains. It is also a necessary complement to the traditional paper-based consumer survey and the potential difference between social media perception and traditional consumer survey results would help regulators better understand the gap in quality of care services from various perspectives.” (line 269-275)
Point 8:Finally, I felt that the manuscript would have benefitted from a close reading in terms of language accuracy as at times the meaning of utterances and claims risked being lost.
Response 8: We are grateful for the suggestion and have revised the utterances and claims of the manuscript suggested by the reviewer.
Round 2
Reviewer 2 Report
Dear Colleagues,
Many thanks for the opportunity to review your revised manuscript. I appreciate the time you have taken to address the points raised in my previous review.
Kind regards